# Verification of Information Thermodynamics in a Trapped Ion System

**DOI:** 10.3390/e24060813

**Published:** 2022-06-11

**Authors:** Lei-Lei Yan, Lv-Yun Wang, Shi-Lei Su, Fei Zhou, Mang Feng

**Affiliations:** 1School of Physics, Zhengzhou University, Zhengzhou 450001, China; 202012132012345@gs.zzu.edu.cn (L.-Y.W.); slsu@zzu.edu.cn (S.-L.S.); 2State Key Laboratory of Magnetic Resonance and Atomic and Molecular Physics, Wuhan Institute of Physics and Mathematics, Innovation Academy of Precision Measurement Science and Technology, Chinese Academy of Sciences, Wuhan 430071, China; zhoufei@wipm.ac.cn; 3Department of Physics, Zhejiang Normal University, Jinhua 321004, China; 4Research Center for Quantum Precision Measurement, Guangzhou Institute of Industry Technology, Guangzhou 511458, China

**Keywords:** trapped ion, quantum information, thermodynamics, entropy production, Maxwell demon, Landauer principle

## Abstract

Information thermodynamics has developed rapidly over past years, and the trapped ions, as a controllable quantum system, have demonstrated feasibility to experimentally verify the theoretical predictions in the information thermodynamics. Here, we address some representative theories of information thermodynamics, such as the quantum Landauer principle, information equality based on the two-point measurement, information-theoretical bound of irreversibility, and speed limit restrained by the entropy production of system, and review their experimental demonstration in the trapped ion system. In these schemes, the typical physical processes, such as the entropy flow, energy transfer, and information flow, build the connection between thermodynamic processes and information variation. We then elucidate the concrete quantum control strategies to simulate these processes by using quantum operators and the decay paths in the trapped-ion system. Based on them, some significantly dynamical processes in the trapped ion system to realize the newly proposed information-thermodynamic models is reviewed. Although only some latest experimental results of information thermodynamics with a single trapped-ion quantum system are reviewed here, we expect to find more exploration in the future with more ions involved in the experimental systems.

## 1. Introduction

As an important subject in modern science, thermodynamics provides a basic understanding for the heat-work natural phenomena, where the most famous are the four laws of thermodynamics, particularly, the second law elaborating the time arrow and entropy production. After discovery of the second law of thermodynamics, a fictive experiment, well known as Maxwell’s demon, was proposed by James Clerk Maxwell to challenge the second law by means of the information of the demon [1,2]. The paradox of Maxwell’s demon revealed, for the first time, the relationship between entropy and information, and elaborated the physical nature of information [3]. Then, Léo Szilárd designed a single particle version of the Maxwell’s demon [4,5], quantizing the work extraction from a thermal reservoir at temperature *T* as $k_{B}T\ln2$ with $k_{B}$ denoting the Boltzmann constant. Today, the information thermodynamics, as an interpretation of the physical nature of information, has extended to different fields, such as the information erasure [6,7], nonequilibrium equality for free energy differences [8,9], irreversibility [10], and quantum fluctuation theory [11].

With the progress of thermodynamics, Maxwell’s demon has gradually formed different modalities. For instance, Maxwell’s demon can be used to design an engine to extract work from quantum measurements [12,13], to realize a steering heat engine in quantum systems [14], and to appear in nonequilibrium systems [15]. Besides, Maxwell’s demon has been produced in different physical systems, such as the superconducting quantum circuits [16], ultracold atoms [17], and solid-state spin degrees of freedom [18]. Meanwhile, the quantum version of Léo Szilárd engine has also been proposed in Ref. [19], while the classical Szilárd engine was experimentally demonstrated in single electron systems [20,21]. So far, Maxwell’s demon connecting thermodynamics and information has seemed to be solved. However, more conundrums still remain, such as quantum information erasure [22,23], thermodynamic uncertainty relation [24,25], and speed limit [26,27,28].

As one of most important theories in information thermodynamics, the Landauer principle stipulates the lower bound of energy dissipation when bit information is erased [3,6,7,29,30]. However, the Landauer bound can only be obtained under the quasi-static condition [31], not for the minimal energy cost [32,33,34,35,36] of the information erasure and for an erasure process [37,38,39]. Due to the experimental difficulties in the single particle manipulation and work measurement, the single particle demonstration of Landauer principle, after many years’ exploration, had been realized by a colloidal particle trapped in a modulated double-well potential [31], by the colloidal particle feedback in a time-dependent virtual potential [40], and by single-bit operations of the nanomagnetic memory bits [41]. The last one of the abovementioned work has not only verified the link between information and thermodynamics through erasure, but also further highlighted the physical limit of irreversible computation. Moreover, the recent development of quantum information has led to more connection of quantum information erasure to different quantum properties, such as the finite heat reservoirs [22,23], squeezed characters [42], quantum fluctuations [43], and zero temperature limit [44]. Due to difficulties in the measurement of heat, coherence, and correlation, the experimental verification of quantum Landauer principle was first implemented until recently by a trapped-ion system [45], where an equality relation, based on the entropy or information conservation of unitary evolution [46], is associated with the energy cost of information erasure in conjunction with the entropy change of the associated quantized environment.

One essential feature of quantum thermodynamics is the fluctuation [8,9,11,47,48,49,50]. Recently, more equality relations have been presented for the fluctuation theorem, such as a new uncertainty related to the fluctuation [51], fluctuation beyond two-point measurements [52], Jarzynski equality in stochastic resetting [53], Jarzynski relation in stochastic dephasing [54], nonequilibrium in the open quantum system [55], and a new interpretation based on the Maxwell’s demon [56]. To experimentally demonstrate the fluctuation theorem, the most important quantity is the measurement of the fluctuated work in a quantum system by an accessible method, for instance, by measuring the characteristic function of work to obtain the average work [57], and by structuring a qubit interferometry to extract the work statistics population [58]. Experimentally, the colloidal particles captured in an optical trap [59] and a time-dependent nonharmonic potential [60] were used to demonstrate the fluctuation theorem and Jarzynski equality; a non-equilibrium feedback Brownian particle [61] has simulated the Szilárd engine and verified the generalized Jarzynski equality [62]. There are other works in this aspect, e.g., RNA folds for free energies [63], the single-molecule pulling experiments [64] to reconstruct the free energy, nonequilibrium measurements [65] to test the Jarzynski equality, and the trapped cold ions system to verify the quantum Jarzynski equality [66,67]. Besides, the Jarzynski equality is related to an information equality [68] by the mutual information between the two-point measurements, which has also been verified by the trapped-ion experiment [69].

Another bridge between information theory and thermodynamics is the information-theoretical bound on irreversibility [70] through the mutual information or relative entropy in the information theory [71,72]. The bounds on irreversibility occur in the open quantum system [73] and thermal relaxation process [74], showing the geometric properties [70,75] of the relative entropy. Due to the negative entropy property of quantum systems induced by coherence [76], entanglement [77], and quantum correlation [78], the information-theoretical bound, quantized by the classical Kullback–Leibler divergence, can be violated in the quantum relaxation process [74,79], which has experimentally witnessed the trapped-ion experiment [79].

As one of the basic characteristics in quantum systems, the uncertainty relation also restricts the non-equilibrium fluctuations of quantum thermodynamic systems [80]. Over the past two years, the thermodynamic uncertainty relations have been proposed in different quantum systems and processes, such as the continuous measurement [81], arbitrary initial states [82], time-dependent driving [83], open quantum systems [84], and the quantum heat engines [85]. Especially, the speed bound of thermodynamic processes exists for the system and is from the dissipation [86], fluctuations [87], and entropy production [88], etc. Then, a new uncertainty relation, linking the speed limit of the processes and entropy production rates [89,90], is put forth, called the dissipation-time uncertainty relation [91], which can be applied to expound the transport precision bound [92], the time measurement limit [93], and the seed bound of molecular motors [94]. In addition, this uncertainty and the speed bound have been verified by a trapped ion quantum system [95,96].

As an admirable platform for the quantum information process, the ion trap systems have been employed as a quantum information processor [97,98] to implement universal quantum logic gates [99,100,101], to demonstrate quantum algorithms [102], and to realize the scalable quantum computation [103,104,105,106,107]. In addition, due to the dual nature of quantum mechanics [108] and thermodynamics [104,109], the ion trap systems can also be applied to create multi-qubit entanglement [110], to verify the Heisenberg uncertainty [111], to prepare nonclassical states [112] and quantum bath [113], and to demonstrate single atom engines [114,115,116]. Moreover, by jointly making use of the electronic state structure, decay channels, and vibrational modes, a single trapped ion can behave as a standard quantum thermodynamic system, verifying or demonstrating some newly developed theories in the information thermodynamics, such as the quantum information erasure [45], the information-theoretical bound of irreversibility [79], the information-theoretic equality [69], and the dissipation-time uncertainty relation in a quantum system [95].

The remaining parts of this paper are structured as follows. In Section 2, we review the quantum operators in a trapped-ion system regarding the interaction, decoherence, and dissipation. Section 3 is arranged to discuss how to define an effective temperature for a two-level system, to build an accessible dissipative channel, and to simulate a heat bath by the vibration modes and decay processes. In Section 4, we focus on the quantum Landauer principle, showing the connection between information and thermodynamics by monitoring the energy dissipation when a bit is erased, and then reviewing its realization in a trapped-ion system. Next, we review the theory and experimental test of information-theoretic equality as well as its Jarzyski equality version in Section 5. Section 6 describes how the relative entropy bounds the irreversibility and how to realize an experimental verification about this bound. In Section 7, we review the entropy production to impose restrictions on the speed limit of physical processes, and then discuss how to produce stochastic trajectories in a practical quantum system. Finally, we offer a brief conclusion.

## 2. The Operators in Ion Trap

When the ion confined in the ion trap is cooled down to the vibrational ground state, the system can behave as a two-level system coupled to a harmonic vibration. Quantum operations based on this model can be performed in such a system. In this section, we introduce the qubit system encoded by the internal electronic states of the ion and create a general quantum operation between the internal states and vibrational states using lasers.

### 2.1. Interaction between Lasers and a Single Ion

The Hamiltonian of an ion experiencing the travelling wave field of laser can be expressed as (ℏ=1)
(1)$$H=\omega_{k}a^{†}a+\frac{\omega_{z}}{2}\sigma_{z}+\frac{\Omega }{2}\left[\sigma_{+}e^{i\eta (a+a^{†})}e^{i(\phi -\omega_{l}t)}+h.c.\right]$$
where *a* ($a^{†}$) is the annihilation (creation) operator of the vibration with the vibrational frequency $\omega_{k}$, $\sigma_{-}$ ($\sigma_{+}$) is the lowing (raising) operator of the two-level system encoded by two internal electronic states, $\omega_{z}$ denotes the corresponding resonance frequency, and Ω (ϕ and $\omega_{l}$) is the Rabi frequency (phase and frequency) of the laser. The Lamb–Dicke parameter of the laser is given by $\eta =kz_{0}$ with the zero-point amplitude of vibration $z_{0}=\sqrt{\hbar /2m\omega_{k}}$, where *k* denotes the wave number of laser and *m* is the mass of ion. In the interacting picture with respect to $H_{0}=\omega_{k}a^{†}a+\omega_{z}\sigma_{z}/2$, the Hamiltonianin in Equation (1) is rewritten as
(2)$$H=\frac{\Omega }{2}\left[\sigma_{+}e^{i\eta (ae^{-i\omega_{k}t}+a^{†}e^{i\omega_{k}t})}e^{i(\phi -\delta t)}+h.c.\right]$$
with the detuning of laser $\delta =\omega_{l}-\omega_{a}$. In the Lamb–Dicke regime (η≪1), keeping to the first order of η, the Hamiltonian is reduced to
(3)$$H=\frac{\Omega }{2}\sigma_{+}\left[1+i\eta \left(ae^{-i\omega_{k}t}+a^{†}e^{i\omega_{k}t}\right)\right]e^{i(\phi -\delta t)}+h.c.$$ Under the weak coupling condition, i.e., $\omega_{k}\gg \Omega$, Equation (3) is further simplified under the rotating wave approximation.
When δ=0, the Hamiltonian is reduced to
(4)$$H_{c}=\frac{\Omega }{2}\left(\sigma_{+}e^{i\phi }+\sigma_{-}e^{-i\phi }\right)$$
which describes the process of carrier transition where only the internal state of ion is controlled by the laser. Then, based on the Schrödinger’s equation, a general operation, defined as $R^{c}\equiv \exp\left(-iH_{c}t\right)$, is realized by the carrier Hamiltonian, as
(5)$$R_{c}\left(\theta ,\phi \right)=\cos\left(\frac{\theta }{2}\right)I-i\sin\left(\frac{\theta }{2}\right)\left(\sigma_{+}e^{i\phi }+\sigma_{-}e^{-i\phi }\right)$$
where θ=Ωτ with τ denoting the pulse duration, and the vibrational state of ion is decoupled from the two-level system. Utilizing this process, we may achieve a universal single qubit phase gate and the population transition between the internal levels of ion by a single pulse or several pulses with different shapes.When $\delta =-\omega_{k}$, the first red-sideband transition, Hamiltonian is obtained as
(6)$$H_{r}=\frac{\eta \Omega }{2}\left(a\sigma_{+}e^{i\phi }+a^{†}\sigma_{-}e^{-i\phi }\right)$$
which, as a Stokes process [117,118], has the same form as the coupling term in the Jaynes–Cummings (JC) model [119]. In this process, the quantum number is conserved that one phonon increase in the vibrational state of ion is accompanied by a photon transferred from the excited state of the ion to its ground state, reflecting the coupling, induced by the laser, between the internal electron spin state and external vibrational state of the ion. The coupling operation is then realized as
(7)$$\begin{array}{ccc} R_{r}\left(\theta ,\phi \right) & = & \cos\left(\frac{\theta }{2}\sqrt{a^{†}a}\right)\sigma_{-}\sigma_{+}+\cos\left(\frac{\theta }{2}\sqrt{aa^{†}}\right)\sigma_{+}\sigma_{-}-i\frac{\sin(\frac{\theta }{2}\sqrt{aa^{†}})}{\sqrt{aa^{†}}}a\sigma_{+}e^{i\phi }\\ & & -i\frac{\sin(\frac{\theta }{2}\sqrt{a^{†}a})}{\sqrt{a^{†}a}}a^{†}\sigma_{-}e^{-i\phi } \end{array}$$
with the pulse length of the red-sideband laser θ=ηΩτ. One application of this operator is to perform the sideband transfer of the population between the vibrational state and internal state, exchanging information between the two degrees of freedom.When $\delta =\omega_{k}$, it reduces to the first blue-sideband transition Hamiltonian to
(8)$$H_{b}=\frac{\eta \Omega }{2}\left(a^{†}\sigma_{+}e^{i\phi }+a\sigma_{-}e^{-i\phi }\right)$$
which shows an anti-Stokes process [117], and the quantum number is not conserved in this process that the vibrational state and the internal state simultaneously obtain or lose a quantum number. Then, the quantum operation under a duration τ can be realized as
(9)$$\begin{array}{ccc} R_{b}\left(\theta ,\phi \right) & = & \cos\left(\frac{\theta }{2}\sqrt{a^{†}a}\right)\sigma_{+}\sigma_{-}+\cos\left(\frac{\theta }{2}\sqrt{aa^{†}}\right)\sigma_{-}\sigma_{+}-i\frac{\sin(\frac{\theta }{2}\sqrt{a^{†}a})}{\sqrt{a^{†}a}}a^{†}\sigma_{+}e^{i\phi }\\ & & -i\frac{\sin(\frac{\theta }{2}\sqrt{aa^{†}})}{\sqrt{aa^{†}}}a\sigma_{-}e^{-i\phi }. \end{array}$$This operator can be used to construct the blue-sideband Rabi oscillations, and applied to measure the phonon number of the vibrational state.

### 2.2. Decoherence and Dissipation Process

There are mainly three different decoherence processes in the trapped ion system, i.e., the decay and dephasing processes of the internal electronic state induced by the level lifetime and magnetic field noise, respectively, and the other one is the heating effect of the vibrational state, coming from the electric field noise of the trap and the background gas collision. In the trapped-ion system, the dynamic process for the decay from the excited state |1〉 to |0〉 can be described as
(10)$$L_{d}\left[\rho \right]=\frac{\Gamma }{2}\left(2\sigma_{-}\rho \sigma_{+}-\sigma_{+}\sigma_{-}\rho -\rho \sigma_{+}\sigma_{-}\right)$$
with the rising operator $\sigma_{+}=\left|1\rangle \langle 0\right|$ and lowering operator $\sigma_{-}=\left|0\rangle \langle 1\right|$. The decay rate Γ=1/τ with τ denote the lifetime of excited state, and ρ denotes the density matrix of system. If there are many decay channels, the total dynamic process is the summation of them, that is, $L_{d}\left[\rho \right]=\sum_{k}L_{kd}\left[\rho \right]$ with $L_{kd}\left[\rho \right]$ corresponding to the dynamics of *k*th decay channel with the decay rate $\Gamma_{k}$. The dephasing process of a qubit encoded in a two-level system can be written as
(11)$$L_{p}\left[\rho \right]=\kappa \left(\sigma_{z}\rho \sigma_{z}-\rho \right)$$
with the dephasing rate κ. Same as the decay process, a total dephasing effect contains all the possible dephasing channels. Generally, the qubit in a trapped ion is encoded in the hyperfine structure or the metastable states to resist the decoherence. Thus, the lifetime is very long. However, the dephasing effect induced by the magnetic field noise is the prime factor of decoherence. For example, the qubit system of trapped ${}^{40}$Ca${}^{+}$ is encoded in the ground state and metastable state, owning a lifetime about one second while an accessible dephasing time less than one millisecond. In some cases, the large decoherence is indispensable if the task is to obtain a mixed state, where the fast dissipative channel is designed by introducing an excited state to the system.

The dynamics regarding heating effect for the vibrational states can be described by the following Lindblad master equation
(12)$$L_{a}\left[\rho \right]=\frac{[N\left(\omega_{k}\right)+1)\gamma }{2}\left(2a\rho a^{†}-a^{†}a\rho -\rho a^{†}a\right)+\frac{N(\omega_{k})\gamma }{2}\left(2a^{†}\rho a-aa^{†}\rho -\rho aa^{†}\right)$$
where γ is the vibrational decay rate and $N\left(\omega_{k}\right)={\left[\exp\left(\hbar \omega_{k}/k_{B}T\right)-1\right]}^{-1}$ is the Boltzamann distribution with the temperature *T*. In the low temperature limit, i.e., $\hbar \omega_{k}/k_{B}T\gg 1$, $N(\omega_{k})∼0$ and the above equation can be reduced to
(13)$$L_{a}\left[\rho \right]=\frac{\gamma }{2}\left(2a\rho a^{†}-a^{†}a\rho -\rho a^{†}a\right).$$ If the cooling approaches the vibrational ground state, the effective temperature of the ion always satisfies the low temperature condition and an accessible vibrational decay rate is about dozens of Hertz. Thus, the heating is negligible. However, in the high temperature limit, i.e., $\hbar \omega_{k}/k_{B}T\ll 1$, $N(\omega_{k})\gg 1$, the Lindblad master equation can be written as
(14)$$L_{a}\left[\rho \right]=\tilde{\gamma }\left(a\rho a^{†}+a^{†}\rho a-a^{†}a\rho -\rho a^{†}a-\rho \right)$$
with the effective heating rate $\tilde{\gamma }=N\left(\omega_{k}\right)\gamma$. Therefore, the high temperature of the system accompanies the large effective heating effect and worsens the fidelity of quantum operators. In this sense, the heating effect can be used to prepare a thermal state of the vibrational mode. Therefore, the total dynamic process of a single ion, under the decoherence, dissipation, and the Hamiltonian of ion, can be expressed in the Lindblad form as
(15)$$\dot{\rho }=-i\left[H,\rho \right]+L_{d}\left[\rho \right]+L_{p}\left[\rho \right]+L_{a}\left[\rho \right],$$
where *H* denotes the Hamiltonian of system and the commutator [H,ρ]=Hρ−ρH.

## 3. Thermodynamic Process of a Trapped Ion

In a trapped ion system, there are two degrees of freedom, i.e., the two-level system formed by the internal electronic states and vibrational mode. Thus, there are two different types of thermodynamic process created by the dissipation of the energy-level and the thermal vibration of the ion.

### 3.1. An Effective Temperature in Two-Level System

Generally, the temperature of system is the representation of the thermal effect for the particles’ motion. However, the thermal state of a two-level system is formed by the steady state of the system, balanced by two dissipative processes of the internal state. Assume that the decay rates $\Gamma_{0\to 1}$ and $\Gamma_{1\to 0}$ correspond to the dissipative processes, respectively, from the state |0〉 to |1〉 and from the state |1〉 to |0〉. Then, the dynamic process of system can be obtained as
(16)$$\frac{d}{dt}\rho_{11}\left(t\right)=-\Gamma_{1\to 0}\rho_{11}\left(t\right)+\Gamma_{0\to 1}\rho_{00}\left(t\right),$$
and $\rho_{00}=1-\rho_{11}$ with the density matrix elements defined as $\rho_{ij}=\langle i|\rho |j\rangle$. When the system reaches the steady state, the populations in two states keep constant, i.e., ${}_{t\to \infty }d\rho_{11}\left(t\right)/dt=0$. Thus, the populations in the steady states are given by $p_{1}=\Gamma_{0\to 1}/\Gamma$ and $p_{0}=\Gamma_{1\to 0}/\Gamma$ for states |1〉 and |0〉, respectively, with the total decay rate $\Gamma =\Gamma_{1\to 0}+\Gamma_{0\to 1}$.

The inverse temperature β in a two-level system is defined as
(17)$$\beta =\frac{1}{\varepsilon }\ln\frac{p_{0}}{p_{1}}$$
where ε denotes the energy difference between the two energy levels. In the steady state, we have $\beta =\varepsilon^{-1}\ln\Gamma_{1\to 0}/\Gamma_{0\to 1}$. When $\Gamma_{1\to 0}>\Gamma_{0\to 1}$, $p_{0}>p_{1}$ and β>0, and thus the temperature is positive. However, if $\Gamma_{1\to 0}<\Gamma_{0\to 1}$, $p_{0}<p_{1}$ and β<0, we have the negative temperature. The negative temperature means the population in the excited state is higher than that of the ground state. In the driven two-level system, the effective temperature can be established by the steady state through considering the combinatory effect of the laser drive and dissipation. However, it generally has a complex form and is also defined in the interaction representation formed by the coherent states basis.

### 3.2. Dissipative Channel Designed by Different Energy Structures

Unlike adding a dissipative process by the demand of the theoretical model, there are no applicable direct dissipative channels for the experimental system to simulate the necessary dissipative process required in the theoretical method, e.g., to simulate a thermodynamic relaxation process of the quantum system. Therefore, designing controllable dissipative channels is essential to verification of the theories of information thermodynamic in quantum systems. In what follows, we present two methods to construct effective dissipative channels in the two-level system.

The simplest method, as shown in Figure 1a, is to create the dissipative channel by using an excited state to enhance the decay rate from the state |1〉 to |0〉. The Hamiltonian in the interaction picture is $H_{s}=\Omega (|2\rangle \langle 1|+|1\rangle \langle 2|)$, with Ω being the Rabi frequency between |2〉 and |1〉. We assume that the decay rate of exited state is much larger than the Rabi frequency of the laser (Γ≫Ω) and the evolution time is much longer than the time-scale of the excited-state lifetime (t≫1/Γ). Then, the dynamic evolution of the population in the state |1〉 is given by [79]
(18)$$p_{1}\left(t\right)=e^{-\frac{1}{2}t\left(\Gamma -\sqrt{\Gamma^{2}-4\Omega^{2}}\right)}.$$

Keeping the second order expansion of Ω, Equation (18) reduces to $p_{1}\left(t\right)=e^{-\Omega^{2}t/\Gamma }$. Besides, the population in the excited state |2〉 is much less than 1. Therefore, we build a dissipative channel from |1〉 to |0〉 with the effective decay rate $\Gamma_{\mathrm{eff}}=\Omega^{2}/\Gamma$. Except the above two restrictions, i.e., Γ≫Ω and t≫1/Γ, there is also an additional weak coupling restriction for the states |1〉 and |0〉, that is, the Rabi frequency between them should be much smaller than Ω, implying that the dynamics between |1〉 and |0〉 must be slower than that between |1〉 and |2〉. Similarly, an inverse dissipative channel from |0〉 to |1〉 can be obtained by an opposite process, as shown in Figure 1c, where the effective decay rate from |0〉 to |1〉 is also also given as $\Gamma_{\mathrm{eff}}=\Omega^{2}/\Gamma$. Figure 2a compares the analytical results with the numerical results and shows that these results fit almost perfectly when the condition Γ≫Ω is satisfied.

A more complicated energy-level for creating an effective dissipative process is presented in Figure 1b, where an additional energy level is involved, together with an auxiliary laser. Then, the evolution of the population in the state |1〉 can be described as $p_{1}=\exp\left(-\Gamma_{\mathrm{eff}}t\right)$ with the analytical result for an effective decay from state |1〉 to |0〉 as [95]
(19)$$\Gamma_{\mathrm{eff}}=\frac{\Omega_{1}^{2}\Gamma }{\Omega_{2}^{2}}.$$ To make this relation valid, we have to set some restrictions for the dynamic evolution between these states, e.g., the decay rate of an excited state much larger than all other characteristic parameters, and the dynamic process between the two-level system (|0〉 and |1〉) slower than other three processes. Thus, we have the following chain relationship, i.e., $\Gamma \gg \Omega_{2}\gg \Omega_{1}\gg \Gamma_{\mathrm{eff}}$. The results are shown in Figure 2b–d, where the parameters to produce a smaller effective dissipation rate always faultlessly satisfy the chain relationship and the analytical and numerical results perfectly fit. In addition, the larger effective dissipation rate needs larger $\Omega_{1}$ or smaller $\Omega_{2}$ according to Equation (19), otherwise, the chain relationship is not fully satisfied and a small deviation between the analytical and numerical results would appear, as shown in the beginning part of the evolution in Figure 2b,c. Besides, the inverse dissipative processes can be realized by adjusting the laser scheme in Figure 1b, to make the decay from the excited state to |1〉 and the laser irradiation couple |0〉 to |2〉. Furthermore, if there are two decay channels from the excited state |3〉 to both states |0〉 and |1〉 with the decay rates $\Gamma_{0,1}$ and $\Gamma =\Gamma_{0}+\Gamma_{1}$, the solution becomes more complicated and is written as [95],
(20)$$\Gamma_{\mathrm{eff}}=\frac{\Omega_{1}^{2}\Gamma_{1}}{\sqrt{\Omega_{2}^{4}+5\Omega_{1}^{2}\Gamma_{2}^{2}}}$$
with the modified chain condition $\Gamma_{1}\gg \Omega_{2}\gg \Omega_{1}\gg \Gamma_{\mathrm{eff}}$. Specially, in the limit of $\Omega_{2}^{4}\gg \Omega_{1}^{2}\Gamma_{2}^{2}$, we have $\Gamma_{\mathrm{eff}}=\Omega_{1}^{2}\Gamma_{1}/\Omega_{2}^{2}$, the same as that in Equation (19). Generally, the chain conditions restrain a large effective dissipative rate, which should be about three orders of magnitudes smaller than the decay rate of excited state. This leads to a much slower dynamic evolution of the qubit system, such as, for Γ/2π=30 MHz, the accessible effective decay rate is smaller than 50 kHz, implying that the execution time of quantum operations needs to be longer than 20 μs.

### 3.3. Quantum Heat Bath Simulated by the Vibrational Mode

In the quantum region, the vibration of ion is described by a quantum harmonic oscillator that is written as $H_{R}=\hbar \omega_{k}a^{†}a$, where $\omega_{k}$ is the vibrational frequency and *a* ($a^{†}$) is the annihilation (creation) operator of phonons. Then, we have a general form of the density matrix for the vibrational state $\rho_{R}=\sum_{m,n=0}^{\infty }R_{mn}\left|m\rangle \langle n\right|$ with $R_{mn}$ denoting the matrix elements and |n〉 being the *n*th Fock state. In particular, if the ion is in the thermal state, the matrix elements are given by $R_{mn}=\langle m|\rho_{R}|n\rangle =\left(1-e^{-\beta \hbar \omega_{k}}\right)e^{-\beta \hbar \omega_{k}n}\delta_{mn}$ with the inverse temperature $\beta =1/k_{B}T$. Then, the thermal state of the vibration is reduced to $\rho_{R}=\sum_{n=0}^{\infty }p_{n}\left|n\rangle \langle n\right|$, where the thermal probability distribution is defined as $p_{n}={\langle n\rangle }^{n}/{\left(\langle n\rangle +1\right)}^{n+1}$ with the average phonon number $\langle n\rangle =\sum_{n=0}^{\infty }np_{n}=1/\left(e^{\beta \hbar \omega_{k}}-1\right)$. Thus, the average phonon number is employed to define an effective temperature as
(21)$$T=\frac{\hbar \omega_{k}}{k_{B}}\ln^{-1}\left(1+\frac{1}{\langle n\rangle }\right).$$ In the limit of 〈n〉≫1, the temperature is simply written as $T=\hbar \omega_{k}\langle n\rangle /k_{B}$. If 〈n〉≪1, it turns to be $T=-\hbar \omega_{k}/k_{B}\ln\langle n\rangle$.

The first step to implement the further experiment is the ground state cooling of the ions. The quantum heat bath can be simulated by the ions’ vibration, which is obtained only after accomplishing the ground cooling of the ions. Cooling the ion’s vibration to the ground state can be obtained using different cooling schemes, such as the sideband cooling and electromagnetically induced transparency (EIT) cooling. After the cooling is achieved, a lower average phonon number, about 〈n〉∼0.1, is obtained, and the state is not in thermal state any more and the phonon distribution no longer obeys the thermal distribution. In this case, to prepare the vibration mode into a thermal state, we execute a thermalization process by utilizing the environmental thermal effect, which causes a heating of the vibration mode. However, the intrinsic heating rate in an ion trap is very small due to the good isolation of the ion from the environment. Enlarging the heating noise and waiting for a long time are the straightforward ways to obtain appropriate thermal states [45]. Despite some studies for the heating of ions [120], speeding up the thermalization by feasible ideas is still challenging in the simulation of the heat bath using the vibration modes.

To simulate a heat bath by the vibration mode, the thermodynamic quantities, such as heat, entropy produce, and information changes, should be measurable, i.e., the probability distribution $p_{n}$ of phonons in the vibrational mode can be experimentally measured. Provided that the system and reservoir are initially in a joint state $\rho_{SR}=\left|0\rangle \langle 0\right|⊗\sum_{n=0}^{\infty }p_{n}\left|n\rangle \langle n\right|$, under the evolution operator of the blue-sideband Hamiltonian in Equation (9), we are able to measure the population in the state |0〉 of qubit system. Then, the probability to measure the population in the internal state |0〉 of a single trapped ion is obtained as [45,67]
(22)$$P_{0}=\mathrm{Tr}[|0\rangle \langle 0|R_{b}\left(\theta \right)\rho_{SR}R_{b}^{†}\left(\theta \right)]=\frac{1}{2}\sum_{n}p_{n}\left(1-\cos\theta \sqrt{n+1}\right)$$
with the laser phase ϕ=0. Considering experimental imperfections, such as the intensity fluctuation of laser, imperfect state preparation, and detection error, the above equation is amended as [67]
(23)$$P_{0}=\frac{1}{2}\sum_{n}p_{n}\left(1-Ae^{-\gamma t}\cos\theta \sqrt{n+1}\right),$$
with the evolution duration t=θ/ηΩ. By fitting Equation (23) to the experimental data, the parameters $p_{n}$ can be acquired.

### 3.4. A ${}^{40}$Ca${}^{+}$ Atom Confined in the Ion Trap

We take the trapped ${}^{40}$Ca${}^{+}$ ion system [111] as an example. Some internal electronic state energy levels are plotted in Figure 3. When a magnetic field is applied, the energy level is split into some sublevels due to the Zeeman effect. In general, an optical qubit (see the energy levels denoted by the green lines in Figure 3), is encoded in a ground state sublevel $\left|0\rangle =\right|S_{1/2},m=+1/2\rangle$ and a metastable state sublevel $\left|1\rangle =\right|D_{5/2},m=+3/2\rangle$, driven by a narrow linewidth laser with the wavelength 729 nm. The initial state preparation of the qubit is carried out by combining the irradiations of 729 nm, 854 nm, and 866 nm lasers. Then, the detection/readout of population is realized by the electron shelving technique, executed by the 397 nm and 729 nm lasers. The quantum operations of the qubit are implemented by the 729 nm laser, and the coupling between the internal states and vibrational mode is also made by the 729 nm laser. The Doppler cooling process of vibration, implemented by the 397 nm and 866 nm lasers, is executed for about 2.5 ms while the sideband cooling using the 729 nm laser is accomplished by for about 30 ms. Typical quantum operations on the qubit or coupling qubit to the vibrational modes, related to the Rabi frequency of the 729 nm laser, is much shorter than 1 ms. Moreover, the fluorescence detection of the population needs about 10 ms. Therefore, a typical experimental sequence, involving the initialization, preparation, quantum control, and detection, needs a time window of 40 ms. In the following, we mainly review the development for the verification of information thermodynamics based on the ${}^{40}$Ca${}^{+}$ quantum system.

The decay rate of qubit is about 1 Hz, implying negligible dissipation. To create an effective dissipative channel, we have to introduce the sublevels of excited state $|P_{3/2}\rangle$ based on the methods presented in Section 3.2. Practically, there is a small population leakage from $|P_{3/2}\rangle$ to $|D_{3/2}\rangle$, which thus limits the lifetimes of dissipative channels. On the other hand, the dephasing time of the qubit is about 1 ms, and the heating rate of the vibrational mode is about 10 ms; thus, both of them are much longer than most quantum operation times. Therefore, if we need to prepare a mixed state of the qubit or a thermal state of the vibrational mode for a quantum thermodynamic process, the large dephasing or heating rate is a prerequisite. The mixed state of the qubit can be fast prepared by establishing two opposite dissipative processes [69]. However, the thermalization method of the ion is absent, and experimentally it always needs to wait for about 10 ms to produce the thermal state [45]. Alternatively, a quantum reservoir simulated by the harmonic vibration can be designed by producing different couplings between the ion and its environment, based on the absorption of a laser photon and subsequent spontaneous emission [113].

## 4. Landauer Principle under Quantum Condition

Information and thermodynamics were first connected by the well-known Maxwell demon, and then extended to the single particle Szilárd engine situation [4,5,19]. The Landauer principle first elaborated the relation between the energy dissipation and information erasure [6]. As the spring up of quantum information, the energy dissipation of quantum information erasure comes out and becomes a hot issue [22,23].

### 4.1. Maxwell Demon and Information Thermodynamics

The Maxwell demon, as shown in Figure 4, is a famous thought experiment presented by Maxwell in 1871. This hypothetical experiment consists of an intelligent demon and an isolated system coupled to a heat environment at uniform temperature. The demon can create the temperature gradient between the left and right parts of the container by measuring the states of system, then extracts work from the system based on the temperature difference, leading to the violation of the thermodynamic second law.

In actual fact, the isolated condition of system itself is invalid and the genuine isolated system turns out to be the composite system including the demon and system. During the process, the demon first measures the speed of particle, and then records the velocity information into its memory. Next, the demon, depending on the information, controls the window in the middle to be open or closed (see Figure 4). After repeating the cycle for many times, the particles with a large speed or kinetic energy are populated in the left part of the container while the particles with a small speed are transferred to the right part, together with the temperature difference arising and the entropy of the system decreasing. Therefore, the demon uses the system to do work by absorbing the heat from the environment. Due to the information of particles’ speed recorded in the memory of the demon in every cycle, the entropy of the demon is continuously accumulated. Thus, the decrease in the system’s entropy is balanced by the entropy increase in the demon, implying that the total entropy of the composite system never decreases. On the other hand, a demon does not have an infinite amount of memory, and the memory needs to be reset or the information in the memory needs to be erased after some cycles of processing.

To erase the information, the demon needs to contact the environment and power source. In the erasure process, the entropy of the demon flows into the environment, accompanied by the heat produced in the environment. Thus, there is at least the same work consumption in the power source to compensate the heat due to the energy conservation. This process is also called the Landauer Principle [6]. For the thermodynamic process, the particle absorbs the heat from the single environment to do work, and then the power source does work to erase the information in the memory of demon, together with the work converted into the heat of environment. Thus, the overall effect of the energy is that the demon utilizes the power source to do work. In the information flow process, the decreased entropy of particles flows into the demon and turns out to be the information of the demon, and then does work. Furthermore, the demon utilizes the power source to erase its memory, together with the information changing into the entropy and heat of environment. Thus, the overall effect of the information is the entropy cycling among the particles, demon, and environment. In short, the demon, as a bridge, connects information to thermodynamics.

### 4.2. Landauer Principle

Landauer principle [6], first proposed by Rolf Landauer, expounds a minimum amount of energy required to be consumed in deletion of a classical bit of information, implying an irreversibility of logical operations [3,7,36]. For a general logical operation process, as shown in Figure 5, the system is initially in the blank state with zero entropy, then turns into a mixed state with a higher entropy. Coupling the system to an environment with temperature *T*, the system is reset to the blank state, together with the entropy of the system transferred into the environment. After a logical process, the mixed state of system has four possible results, i.e., ‘00’, ‘01’, ‘10’, ‘11’, and without loss of generality, their existing probability can be assumed to be equal. Then, the Shannon entropy of system is written as $S\equiv -\sum_{i,j=0,1}p_{ij}\ln p_{ij}=2\ln2$ with $p_{ij}=1/4$. Thus, erasing the information on one bit will transfer ln2 entropy into the environment, corresponding to a $k_{B}T\ln2$ heat increase in the environment. This means that producing this heat requires at least $k_{B}T\ln2$ energy dissipation from the power source. Thus, the Landauer bound is stated as
(24)$$Q\geq k_{B}T\ln2.$$ In a general form, Landauer bound can be written as $Q\geq k_{B}T\Delta S$, with ΔS being the amount of information erased from the system.

The Landauer Principle has been demonstrated in different systems, such as the trapped colloidal particle [31,40], and the nanoscale magnetic memory bits [41]. A typical information erasure process is presented in Figure 6 that the particle is initially prepared in one of the double wells with equal probability, implying that the initial entropy of the system is $S_{i}=k_{B}\ln2$. Then, by lowering the barrier height and applying a tilting force, the particle is moved into the right-side well no matter whether the particle is initially located in the left- or right-side well, together with a zero entropy owned by the system. The process is finished by increasing the barrier to its initial situation. Thus, the minimum entropy production of this erasure process is $k_{B}\ln2$. If the erasure process is quasi-static, the dissipated heat is equal to the Landauer value. However, if the erasure is accomplished within a short/finite time τ, the heat change is asymptotically close to the Landauer limit as $Q=Q_{L}+A/\tau$ with *A* being a positive constant [31]. When the duration is very short, the heat variation can be expressed as an exponential relaxation $Q=Q_{L}+\left(Ce^{-\tau /\tau_{k}}+1\right)A/\tau$, where the characteristic jumping time (Kramers’ relaxation time) and the experimental timescale are $\tau_{k}=\tau_{0}\exp\left(\Delta U/k_{B}T\right)$ and $\tau_{0}$, respectively, with the height of barrier ΔU and temperature *T*.

### 4.3. Quantum Landauer Principle in Trapped Ion System

#### 4.3.1. An Improved Landauer Principle in the Quantum System

As a typical example, an improved Landauer principle was developed in the quantum system, which is an equality relationship with finite-size corrections [23]. As stated above, Landauer process is supposed to erase or reset the state of a system coupled to a thermal reservoir. Based on this conception and as the comparison with the classical version, the erasure process in the quantum system should satisfy the following four conditions:In the erasure process of quantum information, the system *S* and reservoir *R* should be described by the Hilbert spaces;The reservoir *R* is initially prepared in a thermal state described by $\rho_{R}=\exp\left(-\beta H_{R}\right)/Z$ with the partition function $Z=\mathrm{Tr}[\exp\left(-\beta H_{R}\right)]$, where $H_{R}$ denotes the Hamiltonian operator of the reservoir and β is the corresponding inverse temperature, defined by $\beta =1/k_{B}T$;The system and the reservoir are initially uncorrelated, i.e., in the product state $\rho_{S}⊗\rho_{R}$;The erasure process is executed by a unitary evolution $U_{SR}$, i.e., $\rho_{SR}^{\prime }=U_{SR}\rho_{SR}U_{SR}^{†}$.

The first condition is the necessary intrinsic demand for quantum systems. The second condition can apply to the quantum version of the classical thermal reservoir without involving quantum coherence and quantum correlation, which provides the heat and owns the maximum entropy. In contrast, under the same condition, an extra work can be extracted by means of quantum coherence or quantum correlation [76]. The third condition implies the analogy to classical erasure. In contrast, if the initial state is in the entanglement state, the entanglement would work as the negative entropy to extract work [77]. The last condition restricts the system and reservoir as an isolated composite system that no entropy moves in or out from the composite system during the erasure process.

Under these conditions, an equality form of the Landauer Principle [23] can be obtained as
(25)$$\beta \Delta Q=\Delta S+I\left(S^{\prime }:R^{\prime }\right)+D\left(\rho_{R}^{\prime }\|\rho_{R}\right),$$
where the entropy decrease ΔS and the heat dissipation ΔQ are defined in Figure 7, the mutual information and relative entropy are defined, respectively, as $I\left(S^{\prime }:R^{\prime }\right)=S\left(\rho_{S}^{\prime }\right)+S\left(\rho_{R}^{\prime }\right)-S\left(\rho_{SR}^{\prime }\right)$, $D\left(\rho_{R}^{\prime }\|\rho_{R}\right)=\mathrm{Tr}\left[\rho_{R}^{\prime }\ln\rho_{R}^{\prime }\right]-\mathrm{Tr}\left[\rho_{R}^{\prime }\ln\rho_{R}\right]$ with the von Neumann entropy of a quantum state ρ defined as S(ρ)=−Tr[ρlnρ]. The relation in Equation (25) is relevant to the Landauer bound, βΔQ=ΔS, since both the mutual information and relative entropy are non-negative, i.e., $I(S^{\prime }:R^{\prime })\geq 0$ and $D(\rho_{R}^{\prime }\|\rho_{R})\geq 0$. In the classical quasi-static limit, both the mutual information and relative entropy are in proximity to zero, and the relation returns to the Landauer bound.

#### 4.3.2. Single-Atom Demonstration of the Improved Landauer Principle in the Trapped Ion System

The first experimental verification of the improved Landauer principle was implemented in the trapped-ion system [45]. In this experiment, as shown in Figure 8, the two states of the qubit are encoded in the electronic levels as $|↓\rangle =|S_{1/2},m_{z}=1/2\rangle$ and $|↑\rangle =|D_{5/2},m_{z}=3/2\rangle$, and the reservoir is simulated by the vibrational mode of the ion. The system-reservoir coupling is controlled by the red- or blue-sideband operators regarding the 729-nm laser, and the population readout is carried out by detecting the fluorescence of the 397-nm spontaneous emission. In the erasure process, the qubit at the starting point is prepared in a mixed state $\rho_{S}=(|↑\rangle \langle ↑|+|↓\rangle \langle ↓|)/2$, and the reservoir is prepared in the thermal state with energy $E_{0}$ by the thermalization process through waiting for a period of time. In the intermediate step, the 729-nm laser adjusted to the red sideband drives the unitary evolution of the system and reservoir, accompanied by the decrease in system entropy and the increase in reservoir energy. At the end of the erasure, the system is populated in ∣↓〉 and the reservoir ends at a higher energy level.

The state of the reservoir is measured, as shown in Figure 9a, using the analytical formula Equation (23) to fit the experimental data obtained by measuring the blue-sideband Rabi oscillation of the ion. In Figure 9a, the reservoir is initially prepared in the thermal state with an average phonon number 〈n〉=0.15(2), and this can be verified by the typical Rabi oscillation curve induced by the thermal distribution $p_{n}$, as defined in Section 3.3. After the erasure, the Rabi oscillation behaves very differently from the curve caused by the thermal state, displaying the non-thermal property. Besides, the mutual information, denoting the entanglement produced between the system and reservoir, will appear or disappear during the erasing process. However, the relative entropy of the reservoir always increases with the erasing time, which means the enhancement of the nonclassical property when more heat flows into the reservoir. Thus, although the Landauer bound is hard to reach in the quantum system (see Figure 9b), the equality relationship produced by the quantum Landauer principle in Equation (25), as shown by the inset of Figure 9b, is always satisfied.

The quantum Landauer principle as an extension of the Landauer principle in quantum systems, satisfying the above three constraints, has been verified in the ion system, and the energy dissipation is always larger than the bound of the Landauer principle, i.e., $Q>k_{B}T\Delta S$. Although in general the Landauer bound cannot be violated, there exist possibilities to go beyond the Landauer bound in the quantum systems if the constraint is broken. For example, if the system is entangled, the entanglement, as a negative entropy, would make the last two terms in the left-hand of Equation (25) be negative. Thus, a general quantum Landauer principle, containing entanglement and coherence, would be a significative theme. Thus, further experimental exploration of the general quantum Landauer principle is highly expected.

## 5. Connection between Information-Theoretic Equality and Jarzynski Equality

The Jarzynski relation, relating non-equilibrium quantities to the equilibrium free energy, expresses a remarkably simple equality relation and holds in both classical and quantum regimes [8,9]. It can be understood using the fluctuation theorem [47] under the microscopically reversible and thermostat dynamics. So far, the behavior predicted by the Jarzynski equality has been further confirmed in the ensuing investigations, such as the single-molecule pulling experiment [64], mechanically stretching experiment of a single RNA molecule [65] (see Figure 10a,b), and a colloidal particle system [60]. Besides, several attempts have been made to extend the Jarzynski relation to quantum regime [54,58,66], and also verified by an ion-trap experiment [67] (see Figure 10c). From the quantum perspective, the fluctuations originate from both the thermal and quantum effect, and the amount of work itself, which needs to be quantized, is not observable in quantum thermodynamics [58,66]. Besides, a general quantum mechanical process involving a temporal evolution can be sandwiched by two projective measurements, where the measurement process [68], updating the original state to a new state with information encoded [2], can also be observed as a typical thermodynamics process of information. The updated state is generally out of equilibrium after the measurement, even if the system is initially prepared in an equilibrium state.

### 5.1. The Information-Theoretic Equality based on Two-Point Measurements

As shown in Figure 11, the two-point projective measurement process consists of two-measurement bases and a completely positive trace preserving map. In an information theoretic equality, the mutual information is associated with two measurements $\left\{P_{n}\right\}$ and $\left\{Q_{m}\right\}$ before and after a completely positive trace preserving map $F\left(\cdot \right)\equiv \sum_{i}\Lambda_{i}\left(\cdot \right)\Lambda_{i}^{†}$, defined as [68]
(26)$$I_{nm}=-\ln q_{m}+\ln p_{m|n}$$
where the conditional probability, denoting the relevance of the second measurement to the first measurement outcome, is given by $p_{m|n}=\mathrm{Tr}\left[Q_{m}\sum_{i}\left(\Lambda_{i}P_{n}\Lambda_{i}^{†}\right)\right]$, and the *m*th measurement outcome of $\left\{Q_{m}\right\}$, regardless of the results on the basis $\left\{P_{n}\right\}$, is given by $q_{m}=\mathrm{Tr}\left[Q_{m}\sum_{i}\Lambda_{i}\rho \Lambda_{i}^{†}\right]$. Thus, the mutual information quantizes the entropy difference between the *m*th outcomes of the measurement basis $\left\{Q_{m}\right\}$ with and without the knowledge of the *n*th measurement result of the basis $\left\{P_{n}\right\}$. Then, an information-theoretic equality, regarding the mutual information, is written as
(27)$$\langle e^{-I_{nm}}\rangle :=\sum_{nm}p_{nm}e^{-I_{nm}}=1,$$
where the joint probability is $p_{nm}=p_{m|n}p_{n}=\mathrm{Tr}\left[Q_{m}\sum_{i}\Lambda_{i}\left(P_{n}\rho P_{n}\right)\Lambda_{i}^{†}Q_{m}\right]$ with the probability $p_{n}=\mathrm{Tr}\left[P_{n}\rho \right]$ regarding the measurement basis {P}.

Furthermore, if the system is initially prepared as a Gibbs state, the mutual information can be related to the work 〈W〉 performed by the free energy difference ΔF between the initial and final states as
(28)$$I_{nm}=-\beta \left(W_{nm}-\Delta F\right)$$
where the work is defined as $W_{nm}=E_{n}-E_{m}$ with the eigenenergies $E_{n}$ and $E_{m}$ corresponding to the measurement bases $\left\{P_{n}\right\}$ and $\left\{Q_{m}\right\}$, the free energy F=−lnZ/β with the partition function $Z=\sum_{k}e^{-\beta E_{k}}$ and $E_{k}$ denoting the corresponding eigenenergies of these two bases. Inserting Equation (28) into Equation (27) produces the Jarzynski equality exp(〈W〉)=exp(ΔF), with the average work $\langle W\rangle =\sum p_{nm}W_{nm}$. Furthermore, by using the framework of dynamic Bayesian networks, this theory has also been promoted to more detailed and integral quantum fluctuation theorems in a quantum correlated bipartite thermal system, which can capture quantum correlations and quantum coherence at arbitrary times [52].

### 5.2. Verification of Jarzynski-Related Information-Theoretic Equality

As stated by the theory of information-theoretic equality, there are three main elements, i.e., an applicable initial state, two measurement bases and a CPTP channel. Based on the requirement, using a single qubit, combining the evolution process with the two-point measurement can accomplish this task. Figure 12 sketches the experimental processes based on a single qubit, including three laser pulses for two measurements and a CPTP channel. In the first step, the system is initialized in the state ∣↓〉, and then one eigenstate of the measurement basis is produced by applying a carrier-transition operator $U_{c}\left(\theta_{0},\phi_{0}\right)$ on the state ∣↓〉, i.e., $|\xi \rangle =U_{c}\left(\theta_{0},\phi_{0}\right)|↓\rangle$ corresponding to the state of the system after the measurement of $\left\{P_{n}\right\}$. The CPTP map is then executed by the second carrier-transition operator. In this case, this map is a unitary operation. We mention that it is possible to use a non-unitary evolution to create the CPTP map, such as using a dephasing or decay channel. However, a non-unitary channel may be inconvenient to investigate this equality from both analytical calculation and experimental measurement. Finally, similar to the first step, the second carrier-transition operator combines with the projection measurement on the state ∣↑〉, corresponding to the second measurement in the basis $\left\{Q_{m}\right\}$. The information-theoretic equality is then verified by inserting the data, obtained by this process, into Equation (27).

In the second process of Figure 12, the only difference from the first process is an additional dephasing process involved in the initial state preparation for producing a Gibbs state $\rho_{i}=\exp\left(-\beta H_{i}\right)/Z_{i}$ with the partition function $Z_{i}=\mathrm{Tr}\left[\exp\left(-\beta H_{i}\right)\right]$. Since the dephasing time of the internal levels of ion is shorter than 1 ms, to make the non-diagonal terms of the density matrix disappear, we wait for about 3 ms after the carrier-transition operator $U_{c}\left(\theta_{0},\phi_{0}\right)$ is implemented. In this situation, the Hamiltonians corresponding to $\left\{P_{\pm }\right\}$ and $\left\{Q_{\pm }\right\}$ are given, respectively, by $H_{i}=\sum_{\pm }E_{\pm }^{i}P_{\pm }$ and $H_{f}=\sum_{\pm }E_{\pm }^{f}Q_{\pm }$, where $E_{\pm }^{i,f}$ are the corresponding eigenenergies of $H_{i,f}$ with $|E_{\pm }^{i,f}|=E$ for the qubit system, and the joint probability is obtained as $p_{nm}=\mathrm{Tr}\left[Q_{m}P_{n}\right]e^{-\beta E_{n}^{i}}/Z$ with n,m=± and partition function $Z=e^{-\beta E}+e^{\beta E}$. Then, the verification of the Jarzynski-related information-theoretic equality can be made by inserting the experimental data into Equation (28) [69].

## 6. Information-Theoretical Bound of Irreversibility in Quantum Systems

A typical thermodynamic phenomenon of entropy production [10] is the irreversibility [70,73,74,75]. Generally, the irreversibility is embodied by the second law of thermodynamics, and bounded by the principle of entropy increase. However, the bound imposed by the second law is loose and trivial, and thereby some tighter bounds about irreversibility have been presented, such as, the geometrical bounds on irreversibility [73,75] and the information-theoretical bound [70]. Furthermore, the information-theoretical bound derived in the classical standard framework of stochastic thermodynamics was proven to be violated due to coherence involved in quantum systems [79], and thus theoretical exploration was generalized to quantum systems [74].

### 6.1. Classical Information-Theoretical Bound of Irreversibility

The characterization of thermodynamic irreversibility can be described by the entropy production, which can be mathematically expressed by the Kulback–Leibler divergence [70]. As a simple application, Ref. [70] provides an information-theoretical bound on entropy production in thermal relaxation processes, which makes the thermal relaxation processes bounded in terms of the information geometry. In the standard framework of classical stochastic thermodynamics, a probability distribution $p_{i}\left(t\right)$ (for *i*th state) of the system evolves as ${\dot{p}}_{i}\left(t\right)=\sum_{k}w_{ik}p_{k}\left(t\right)$ with the transitional matrix element $w_{ik}$ from the *k*th state to *i*th state satisfying the detailed-balance condition $w_{ik}\exp\left(-\beta E_{k}\right)=w_{ki}\exp\left(-\beta E_{i}\right)$, where β and $E_{i}$ are the inverse temperature and instantaneous energy of the *i*th state, respectively. Then, the total entropy production at the final time τ of the evolution process can be integrated as $\sigma_{\tau }=\int_{0}^{\tau }\dot{\sigma }\left(t\right)dt$, where the entropy production rate satisfies $\dot{\sigma }\left(t\right)={\dot{\sigma }}_{\mathrm{sys}}\left(t\right)+{\dot{\sigma }}_{\mathrm{bath}}\left(t\right)$ with the entropy production rates ${\dot{\sigma }}_{\mathrm{sys}}\left(t\right)=dH\left(p\right)/dt$ and ${\dot{\sigma }}_{\mathrm{bath}}\left(t\right)=-\beta \sum_{i}E_{i}{\dot{p}}_{i}$, and the Shannon entropy $H\left(p\right)=-\sum_{i}p_{i}\ln p_{i}$. Then, the entropy production rate can be expressed by the variational expression as [70,71]
(29)$$\dot{\sigma }\left(t\right)=\max_{q}\left[-\dot{D}\left(p\left(t\right)\|q\left(t\right)\right)\right],$$
with the Kullback–Leibler relative entropy $D\left(p\left(t\right)\|q\left(t\right)\right)=\sum_{i}p_{i}\ln\left(p_{i}/q_{i}\right)$[70] and q(t) being the probability distribution of the inverse process for p(t). As a consequence of the variational principle, Equation (29) is maximized when the inverse process is in the equilibrium distribution with instantaneous energies, i.e., $q\left(t\right)=p^{\mathrm{eq}}$. By setting q(0)=p(τ) and integrating Equation (29) from t=0 to τ, we have
(30)$$\sigma_{\tau /2}\geq D\left(p\left(0\right)\|p\left(\tau \right)\right),$$
which states a stronger constraint on the entropy production than the standard second law of thermodynamics due to non-negativity of the relative entropy. As an example, the stochastic thermodynamics based on a classical three-level system, as shown in Figure 13a, demonstrated that this inequality bound works for the whole time region, where a significant reduction of the difference between the entropy production and the relative entropy is induced by the local thermodynamic equilibrium of the system. In addition, the prerequisites of Equation (30) are: the time-independent transition matrix, no coherence in the initial state of system, no entanglement between the system and bath, and no drive for the system or bath.

Combining the standard expression of entropy production $\sigma_{\tau }=D\left(p\left(0\right)\|p^{\mathrm{eq}}\right)-D\left(p\left(\tau \right)\|p^{\mathrm{eq}}\right)$ and Equation (30), a nontrivial information geometry constraint can be obtained as
(31)$$D\left(p\left(0\right)\|p^{\mathrm{eq}}\right)\geq D\left(p\left(0\right)\|p\left(\tau \right)\right)+D\left(p\left(\tau \right)\|p^{\mathrm{eq}}\right),$$
which locates at the point forming the obtuse angle, as shown in Figure 13b, making the relaxation path confined inside the circle with a diameter determined by p(0) and $p^{\mathrm{eq}}$. This bound is stronger than the conventional second law of thermodynamics defined by the standard expression of $\sigma_{\tau }$. Moreover, if defining the average entropy production rate $\bar{\sigma }=\sigma_{\tau }/\tau$, we find that the speed limit of the stochastic thermodynamics process [89] can be stated as $\tau \geq D\left(p\left(0\right)\|p\left(\tau \right)\right)/\bar{\sigma }$, implying a large entropy production rate required for a short relaxation time [91]. This result has been verified experimentally [95].

### 6.2. Quantum Information-Theoretical Bound of Irreversibility

#### 6.2.1. Violation of Classical Information-Theoretical Bound in a Two-Level System

In quantum systems, the first violation of Equation (30), as exemplified by a driven two-level system, was observed experimentally in Ref. [79]. To view this violation, we consider the balance parameter $\Upsilon \left(t\right)=\sigma_{\left[0,t\right]}-D\left(\rho_{sys}\left(0\right)∥\rho_{sys}\left(t\right)\right)$ and see whether it is smaller than zero or not: If Υ(t)<0, the bound is violated. We focus on the simplest model consisting of a drive and a decay channel, described by the Lindblad master equation $\dot{\rho }=-i\left[H,\rho \right]+\Gamma \left(2\sigma_{-}\rho \sigma_{+}-\sigma_{+}\sigma_{-}\rho -\rho \sigma_{+}\sigma_{-}\right)/2$ with the Hamiltonian $H=\Omega \sigma_{x}/2$ and the decay rate Γ. When t→∞, the system reaches the steady state with the eigenspectrum decomposition representation as $\rho_{s}=\lambda_{-}{|\phi \rangle }_{-}\langle \phi |+\lambda_{+}{|\phi \rangle }_{+}\langle \phi |$ with $\lambda_{-}>\lambda_{+}$, that the effective inverse temperature and Gibbs state can be obtained by the methods given in Section 3.3. As shown in Figure 14, in the ultra-large DDR, Υ(t) is given by $\Upsilon \left(t\right)=-\Omega^{2}t^{2}$ in the short-time limit $t\ll \Omega^{-1}$, which thus demonstrates the violation of the information-theoretical bound of irreversibility. Besides, we may also find the revival of the information-theoretical bound as the evolution of the relaxation process, see Figure 14b, which originates from the competition between coherence and relaxation regarding the system. However, realizing the ultra-large DDR experimentally is very difficult and not accessible for the ${}^{40}$Ca${}^{+}$ system [79].

#### 6.2.2. Coherence as a Source to Invalidate the Bound of Irreversibility

To witness the violation of the bound, we may choose another method using coherence as the source of negative entropy. An arbitrary quantum state of the two-level quantum system can be expressed as $\rho_{i}=\left(I+r\cdot \sigma \right)/2$ with the Bloch vector r=(x,y,z) and Pauli operators $\sigma =(\sigma_{x},\sigma_{y},\sigma_{z})$. Thus, the initial entropies of the system and bath are obtained as
(32)$$S_{sys}=-\sum_{k=\pm }\lambda_{k}\ln\lambda_{k},S_{bath}=-\beta \sum_{k=\pm }E_{\pm }\lambda_{\pm },$$
where the eigenvectors and eigenenergies of the initial state are given, respectively, as $\lambda_{\pm }=\left(1\pm |r|\right)/2$ and $E_{\pm }=\left[(|r|\pm z\right)E_{2}+\left(|r|\mp z\right)E_{1}]/2|r|$ with the energy difference $\Delta E=z(E_{2}-E_{1})/|r|$. Nevertheless, the analytical solution for the thermal relaxation process from an arbitrary initial state is intractable. Instead, the numerical results for different initial states (see Figure 15a) show that the bound violation would possibly occur since the system evolves away from the initially mixed state, and the total entropy production, being of negative values in some situations, will be smaller than the Kullback–Leibler divergence, as witnessed in Figure 15b.

#### 6.2.3. A Bound for Irreversibility in a Qunatum Open System

The thermal relaxation process in a Markovian open quantum system is then described by Ref. [74], where, based on the Lindblad master equation and unitary evolution operator, a lower bound of the entropy production, as shown in Figure 16a, is stated as [74]
(33)$$\sigma_{\tau }\geq D_{M}\left[U_{\tau }\rho \left(0\right)U_{\tau }^{†}\|\rho \left(\tau \right)\right],$$
where the unitary $U_{\tau }=\exp\left(-iHt\right)$ and projection measurement relative entropy $D_{M}\left[\sigma \|\rho \right]=D\left(\sigma \|\sum_{n}\Pi_{n}\sigma \Pi_{n}\right)$ with $\Pi_{n}$ denotes the eigenbasis of ρ. This bound also strengthens the Clausius inequality in the conventional second law of thermodynamics and is geometrical, tight, and can be saturated. In addition, it also elaborates a quantum speed limit by $\tau \geq D_{M}\left[U_{\tau }\rho \left(0\right)U_{\tau }^{†}\right\|\rho \left(\tau \right)/\bar{\sigma }$ with the average entropy production rate $\bar{\sigma }=\sigma_{\tau }/\tau$, implying that a fast state transformation requires a high dissipation rate [75,89,91,95]. In the classical limit, no coherence is involved and the lower bound reduces exactly to the bound in Ref. [70]. Furthermore, considering to reset the qubit from a maximally mixed state to its lower state, the minimal heat dissipation, related to the Landauer principle, is
(34)$$\Delta S_{\mathrm{env}}\geq -\Delta S_{\mathrm{sys}}-\frac{1}{2}\ln\left[4\delta \left(1-\delta \right)\right]$$
with δ=〈0|ρ|0〉. This indicates that a higher resetting precision needs more heat dissipated into the environment.

Consider an autonomous thermal machine [50,93] with three levels as a simple example (see Figure 16b), which is driven by two reservoirs at different inverse temperatures $\beta_{1}>\beta_{2}$. The Hamiltonian of the system, given by $H=\omega_{1}|\epsilon_{A}\rangle \langle \epsilon_{A}|+\omega_{2}|\epsilon_{B}\rangle \langle \epsilon_{B}|$, has four jumping operators describing the translations $|\epsilon_{A}\rangle ↔|\epsilon_{g}\rangle$ and $|\epsilon_{g}\rangle ↔|\epsilon_{B}\rangle$, respectively. One example to verify Equation (33) is to set the initial state of system as a pure state ρ(0)=|ϕ〉〈ϕ| with $|\phi \rangle =\sqrt{10-\kappa }|\epsilon_{g}\rangle /3+\sqrt{\kappa -1}|\epsilon_{B}\rangle /3$ and the parameter $\kappa =\beta_{1}/\beta_{2}$. If κ=1; the system relaxes to the equilibrium Gibbs state, while if κ>1, coherence emerges in the initial state and the finial state will be in a nonequilibrium steady state with an entropy flow existing between the two baths. To verify experimentally the information-theory bound of Equation (33) by the three-level system, we have to construct six decay channels to simulate all the jumping operators that the decay channels can be established by utilizing the method presented in Section 3.2. However, creating six decay channels is intractable or inaccessible for the trapped calcium ion system [95].

## 7. Thermodynamic Speed Limit Restrained by the Entropy Production

The thermodynamic speed limit has been extensively investigated by the information-theoretical bound of irreversibility [24,75,89], the thermodynamic uncertainty [24], and quantum speed limits [26,27,28]. Recently, this study produced a new inequality, called the dissipation-time uncertainty relation [91]. This new uncertainty relation was recently checked in a trapped-ion experiment [95].

### 7.1. Dissipation-Time Uncertainty Relation

The dissipation-time uncertainty relation states that the rate of a physical process performed in a stochastic system is bound by the entropy production rate. Specifically, under the assumption of the reverse process of a nonequilibrium system to be tardy and rare, the dissipation-time uncertainty principle is given by
(35)$$\langle {\dot{S}}_{e}\rangle T\geq k_{B}$$
where $\langle {\dot{S}}_{e}\rangle$ and T correspond, respectively, to the mean entropy flow and mean existence time of a process. One typical model to witness the dissipation-time uncertainty is the overdamped particle transport in one spatial dimension, as shown in Figure 17a, described by the dynamics $\dot{x}=-dU/dx+\sqrt{2/\beta }\xi$ with ξ denoting the zero-mean Gaussian white noise of the unit variance. In the stationary regime, the process is described as $O=\int_{0}^{t}\dot{x}\left(\tau \right)d\tau$ with the specific domain D=[2πN,∞), and the trajectories of the reverse process, described by $\tilde{O}=-2\pi N$, are much rarer for large *N* or β. This model can be used to describe a wealth of transport processes between molecular motors and Josephson junctions [94].

Another model to witness the dissipation-time uncertainty is the energy transfer between two heat baths with different inverse temperatures $\beta_{1,2}$ as in Figure 17b and intermediated by a two-level system. The process describing an energy *E* transferring into the cold bath in a fixed time δ, is given by
(36)$$O=\epsilon \int_{0}^{\delta }\left(\frac{dN_{e\to g}^{c}}{d\tau }-\frac{dN_{g\to e}^{c}}{d\tau }\right)d\tau ,\;D=\left[E,\infty \right],$$
where $\epsilon =\epsilon_{e}-\epsilon_{g}$, and the Poisson processes $dN_{i\to j}^{\nu }$ correspond to the Markovian jumps between these two states. The reverse process in this model, i.e., extracting an energy larger than *E* in the time duration δ from the cold reservoir and then injecting into the hot bath, is negligible. Moreover, the entropy flow in this model can be analytically solved as ${\dot{S}}_{e}=k_{B}\epsilon \left(\beta_{h}-\beta_{c}\right)\left(e^{\beta_{h}\epsilon /2}-e^{\beta_{c}\epsilon /2}\right)/\left[1+e^{(\beta_{h}+\beta_{c})\epsilon /2}\right]$. Based on this model, an experiment in a trapped-ion system has been implemented, verifying the dissipation-time uncertainty in Equation (35).

### 7.2. Verification of Speed Limit in the Trapped-Ion System

#### 7.2.1. Stochastic Trajectories of the Dissipative Processes in a Trapped-Ion System

To demonstrate the dissipation-time uncertainty experimentally, we have to first set up the two-level dissipative model. Practically, producing two heat baths with different temperatures and simultaneously coupling them to a two-level system is troublesome and unrealistic. Instead, an accessible method is to simulate all the possible stochastic processes by structuring four dissipative channels based on the energy-level structure of ion. The channels in Figure 18 originate from the method for the effective decay channels presented in Section 3.2.

After the decay channels are built, the next step is to produce stochastic trajectories of the dissipative process. Due to the experimental drawback, such as the cross talk of different dissipative channels, population leakages, and fluorescence detection, we have developed a discretization-repetition ensemble average method by applying the quantum jump theory [122] to produce stochastic evolution trajectories in the data analysis [95]. By separating the whole duration of the evolution into *n* pieces, we quantify the energy transfer in each piece, and connect all the durations to construct a stochastic evolution trajectory over the time period nτ (Figure 19a). Then, the energy transfer process described by Equation (36) is produced by the evolution trajectory. Since each of the stochastic processes is independent from each other, an equilibrium process is then obtained by averaging all the possible evolution paths.

#### 7.2.2. Experimental Verification of Speed Limit

Based on the stochastic evolution trajectories, the survival probabilities of the processes, representing the probability of the trajectory’s first passage time τ longer than the threshold time *t*, can be obtained by the ensemble average of all the possible trajectories. Then, a rare reverse process can be experimentally confirmed by the approximately exponential decays of the survival probabilities with respect to the evolution time. Besides, as an auxiliary substance of the dissipation-time uncertainty, the entropy flux also sets an upper bound on the time-averaged rate $\bar{r}\left(t\right)$ of the process as
(37)$$\langle {\dot{S}}_{e}\rangle /k_{B}\geq \bar{r}\left(t\right).$$ Violation of the entropy flux bound can be experimentally witnessed during the non-stationary dynamic process of the system. However, no violation is found from the experimental observation presented in Figure 19b. Nevertheless, if coherence is involved in dissipative process, the dissipation-time uncertainty presented in Equation (35) might be modified. Therefore, a more general dissipation-time uncertainty relation, containing coherence and other quantum properties, is expected for describing quantum relaxation and nonequilibrium processes.

## 8. Conclusions

In this paper, we have addressed several informational issues connecting the thermodynamics to some experimental works for information thermodynamics in a trapped-ion system. We have briefly reviewed the quantum and thermodynamic properties of a trapped-ion quantum system, and elaborated how to construct decay channels as well as how to create dissipative processes to simulate heat baths. We have also reviewed some recently proposed and experimentally accessible information thermodynamic theories, such as the quantum Landauer principle, the information-theoretical bound of irreversibility, and the speed limit bounded by entropy production. Moreover, we have mentioned the conditions to test them experimentally and then reviewed the relevant experimental demonstration in a trapped-ion system. We anticipate that more experimental works would appear in the flourishing field of information thermodynamics, with not only a single trapped ion, but also a multi-ion system, as well as other quantum qubit systems, such as the superconducting quantum circuits [16,123,124], solid-state spins [18], and ultra-cold atoms [17,125,126], and the microscopic [64] or mesoscopically thermodynamic system [31,40,41,60,61,63,65].

## Figures and Tables

**Figure 1 entropy-24-00813-f001:**
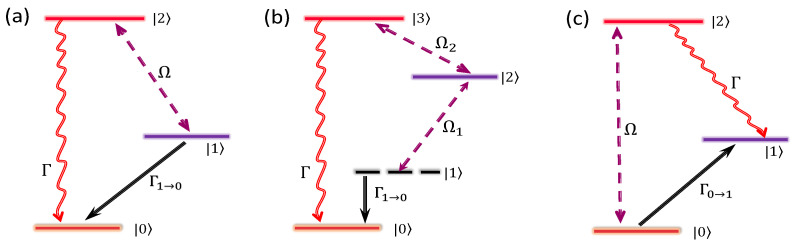
(**a**) Three-level systems for building an effective dissipative channel between two ground states |0〉 and |1〉 by an excited state |2〉 where a laser resonantly couples the ground state |1〉 to the state |2〉 with a Rabi frequency Ω, and then the population decays to the ground state |0〉 from the short-lived excited state. (**b**) Four-level systems for effective dissipative channels between two ground states |0〉 and |1〉 via an intermediate state |2〉 and an excited state |3〉, where two lasers are employed to resonantly couple the ground state |1〉 to the excited state |3〉 via the intermediate state |2〉 with the corresponding Rabi frequency $\Omega_{1,2}$. (**c**) The inverse process to build the effective dissipative channel from the state |0〉 to the state |1〉 based on the three-level system, with respect to (**a**), for an opposite drive-decay process.

**Figure 2 entropy-24-00813-f002:**
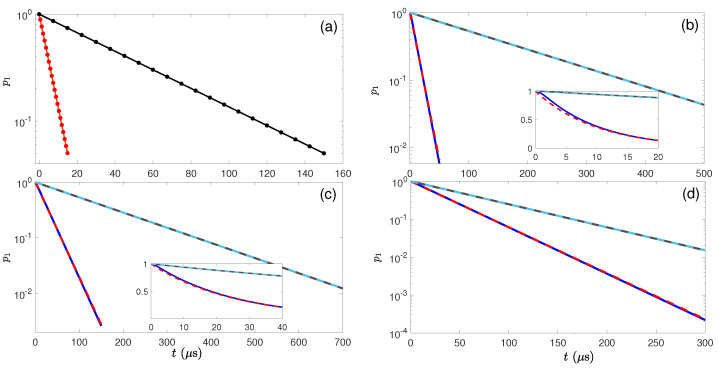
Time evolution of population in the state |1〉 due to the effective dissipative channels constructed in (**a**) a three-level system and (**b**–**d**) a four-level system presented in Figure 1. The dots or dashed curves in each panel are the analytical results obtained by the equation $p_{1}=\exp\left(-\Gamma_{\mathrm{eff}}t\right)$, and the solid curves are from the numerical calculation of the Lindblad master equation. The Rabi frequency in (**a**) is set as Ω/2π=560,178 kHz for the red and black curves, respectively. In (**b**,**c**), two Rabi frequencies of the black and blue curves correspond to $\Omega_{1}/2\pi =20,\;80$ kHz and $\Omega_{2}/2\pi =2$ MHz, and $\Omega_{1}/2\pi =50$ kHz and $\Omega_{2}/2\pi =5,\;2$ MHz. In (d), the three parameters are $\Omega_{1}/2\pi =50$ kHz, $\Omega_{2}/2\pi =3$ MHz, and Γ/2π=8 MHz (black curve) and 16 MHz (blue curve). The decay rate is set as Γ/2π=10 MHz in (**a**–**c**).

**Figure 3 entropy-24-00813-f003:**
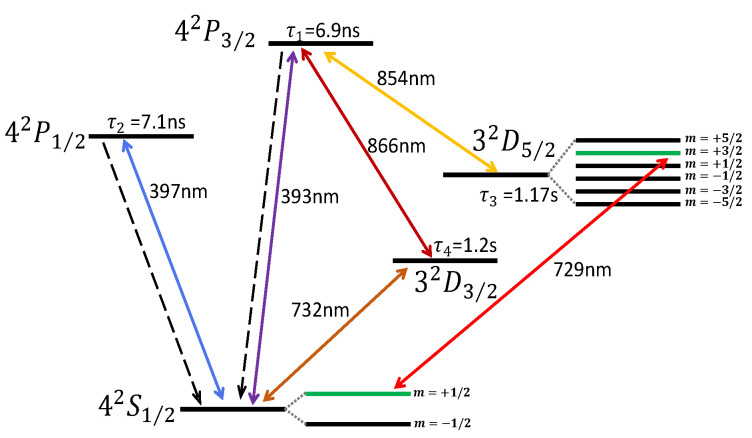
Schematic of the level structure, where one ground state |S〉, two excited states |P〉, and two metastable states |D〉 are shown. A general laser scheme employed experimentally is presented, where the double sided arrows denote the corresponding coupling lasers with the wavelengths labeled. Next to the energy levels, the lifetimes of the excited states and the metastable states are labeled.

**Figure 4 entropy-24-00813-f004:**
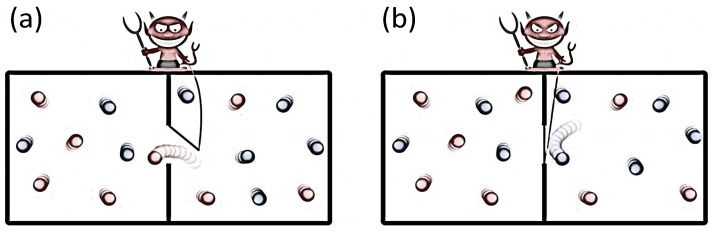
Schematic of Maxwell demon, adapted from [121], where (**a**) the demon, regarding an intelligent device, can measure the speed or kinetic energy of particle around the window in the middle of container, and then determines whether the particle can pass through the window to the other side or not. (**b**) After a period of time, the left part of container owns a higher temperature than the right part, and thus the entropy of system decreases.

**Figure 5 entropy-24-00813-f005:**
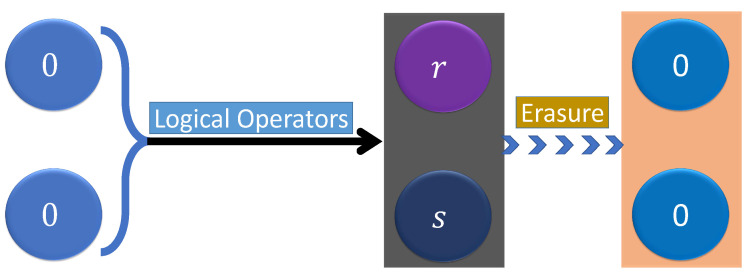
A scheme to illustrate the erasure process where two bits in the state ’0’ undergo some logical operations with the terminal states in ‘r’ and ‘s’, respectively. After the erasure, the states of the bits are reset into their initial states.

**Figure 6 entropy-24-00813-f006:**
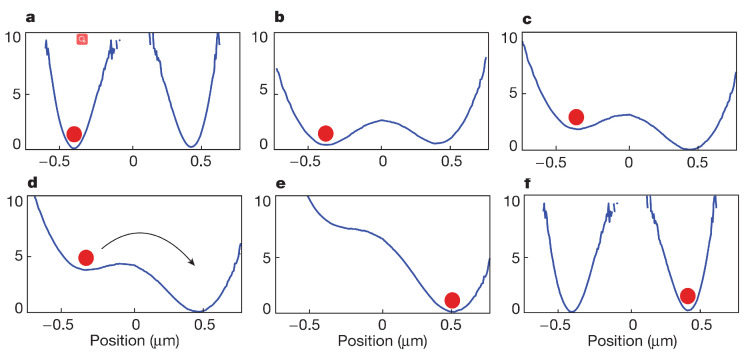
The erasure protocol for a single colloidal particle trapped in a modulated double-well potential, adopted from [31], where (**a**,**b**) a one bit information stored in a bistable potential is erased by first lowering the central barrier and (**c**,**d**) applying a tilting force to drive the particle from the left-hand well to the right-hand well. (**e**,**f**) Independent on the initial state of particle, the final state of particle is always in the right-hand well.

**Figure 7 entropy-24-00813-f007:**
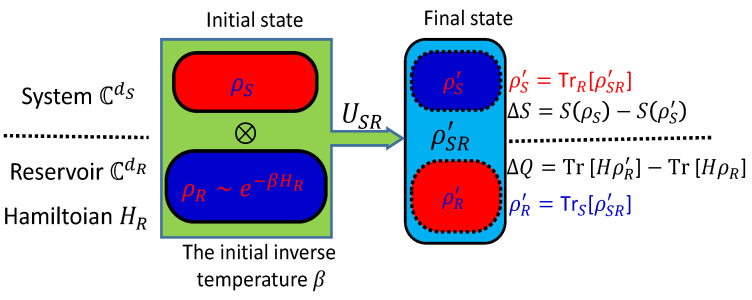
The formalization of the improved Landauer principle, adopted from [23], where the system *S* and a thermal reservoir *R* are initially prepared in an uncorrelated state $\rho_{SR}=\rho_{S}⊗\rho_{R}$ with the thermal state $\rho_{R}=e^{H_{R}}/\mathrm{Tr}\left[e^{H_{R}}\right]$, and then an erasure operator is implemented by a unitary evolution $U_{SR}$, with the final state as $\rho_{SR}^{\prime }=U_{SR}\rho_{SR}U_{SR}^{†}$. The final states of the system and reservoir are replaced by the marginal state as $\rho_{S}^{\prime }={\mathrm{Tr}}_{R}\left[\rho_{SR}^{\prime }\right]$ and $\rho_{R}^{\prime }={\mathrm{Tr}}_{S}\left[\rho_{SR}^{\prime }\right]$, where the Landauer Principle relates the entropy decrease ΔS of system to the heat ΔQ dissipated to the reservoir with βΔQ≥ΔS.

**Figure 8 entropy-24-00813-f008:**
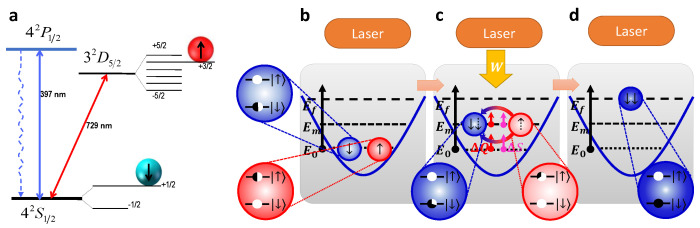
A single trapped-ion system for verification of the improved Landauer principle. (**a**) The energy-level scheme, adopted from [45], and (**b**–**d**). Erasure process of the information encoded in the qubit system.

**Figure 9 entropy-24-00813-f009:**
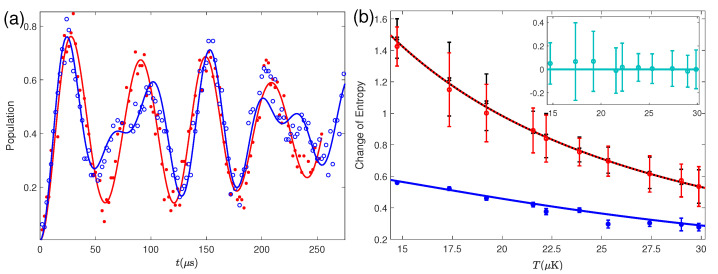
(**a**) Blue-sideband Rabi oscillations before (red curve and data) and after (blue curve and data) the erasure for a waiting time τ=55 ms. (**b**) Test of the improved Landauer principle at different reservoir temperatures, where the black curve (crosses), red curve (circlets), and blue curve (dots) are corresponding to the numerical (experimental) results of $\Delta Q/k_{B}T$ with $\Delta Q=Q_{0}\left(\langle n\rangle -{\langle n\rangle }_{0}\right)$, $\Delta S+I\left(S^{\prime }:R^{\prime }\right)+D\left(\rho_{R}^{\prime }∥\rho_{R}\right)$, and ΔS with the reservoir temperature defined by the initial average phonon number ${\langle n\rangle }_{0}$ as $T=T_{0}/\ln\left(1+1/{\langle n\rangle }_{0}\right)$ with $T_{0}=\hbar \omega_{z}/k_{B}=48.5\;\mu$K. Inset of (**b**) shows the difference between the left-side ($\Delta Q/k_{B}T$) and right-side ($\Delta S+I\left(S^{\prime }:R^{\prime }\right)+D\left(\rho_{R}^{\prime }∥\rho_{R}\right)$) of Equation (25).

**Figure 10 entropy-24-00813-f010:**
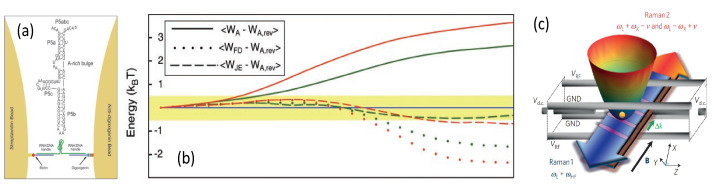
(**a**) Scheme to verify the Jarzynski Equality by the RNA molecules; (**b**) Verification of Jarzynski equality by the dashed lines within the experimental errors. (**a**,**b**) are adopted from [65]. (**c**) Schematic of the trapped ${}^{171}$Yb${}^{+}$ system to verify the quantum Jarzynski equality in an isolated system [67].

**Figure 11 entropy-24-00813-f011:**
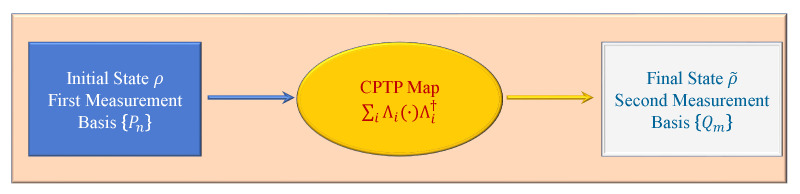
Scheme of the two-point measurement process with CPTP denoting the completely positive trace preserving.

**Figure 12 entropy-24-00813-f012:**
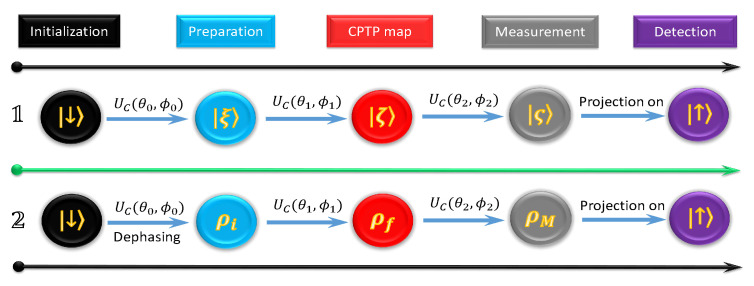
The time sequence to verify the information-theoretic equality and the Jarzynski equality, adopted from [69], where the two states of the qubit are defined as $|↓\rangle =|S_{1/2},m_{z}=1/2\rangle$ and $|↑\rangle =|D_{5/2},m_{z}=3/2\rangle$, respectively. All the evolution processes, implemented by the carrier-transition operator $U_{c}\left(\theta_{k},\phi_{k}\right)$ with the duration $\theta_{k}$ and phase $\phi_{k}$ (k=0,1,2), are presented by the same form as in Equation (Equation 5).

**Figure 13 entropy-24-00813-f013:**
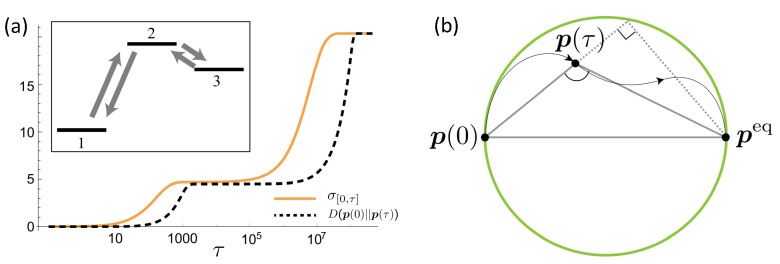
(**a**) Numerical simulation of a classical three-level system for demonstrating Equation (30), where the inset denotes the stochastic transitions between these states. (**b**) Schematic for the constraint presented in Equation (29). The figures are adopted from [70].

**Figure 14 entropy-24-00813-f014:**
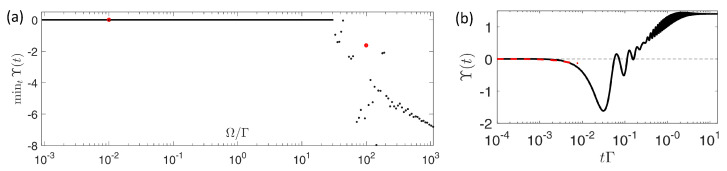
(**a**) Minimum values of the balance parameter Υ(t) for different drive-to-decay ratio (DDR) and (**b**) the balance parameter Υ(t) in the ultra-large DDR case where the black solid (red dashed) curve denotes the numerical (analytical) time evolution of Υ(t). The figures are adopted from [79].

**Figure 15 entropy-24-00813-f015:**
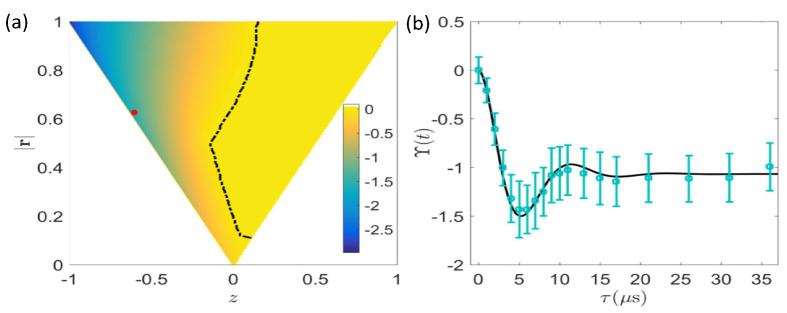
(**a**) Minimum values of the balance parameter for different initial states for the DDR Ω/Γ=1.8, where the dashed curve is the zero line of $\min_{t}\Upsilon \left(t\right)$. (**b**) Experimental result for the time evolution of the balance parameter for the initial state denoted by the red dot in (**a**). The figures are adopted from [79].

**Figure 16 entropy-24-00813-f016:**
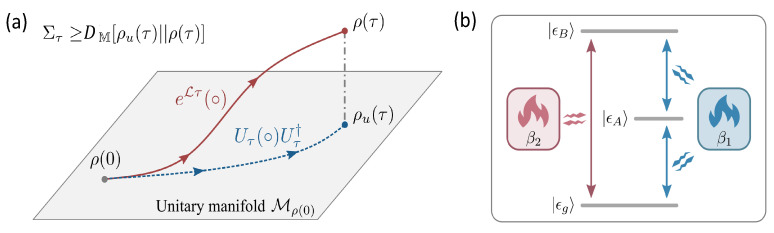
(**a**) Schematic to show the information-theoretical bounds under the Lindblad (solid line) and unitary (dashed line) dynamics, respectively, where the irreversible entropy production $\sigma_{\tau }$ is bounded from below by the information-theoretical distance $D_{M}\left[\rho_{u}\left(\tau \right)\|\rho \left(\tau \right)\right]$. (**b**) Schematic diagram of the two-reservoir machine where the three levels are $\{|\epsilon_{g}\rangle ,|\epsilon_{A}\rangle ,|\epsilon_{B}\rangle \}$ with the energy $0,\omega_{1},\omega_{2}$, respectively. The figures are adopted from [79].

**Figure 17 entropy-24-00813-f017:**
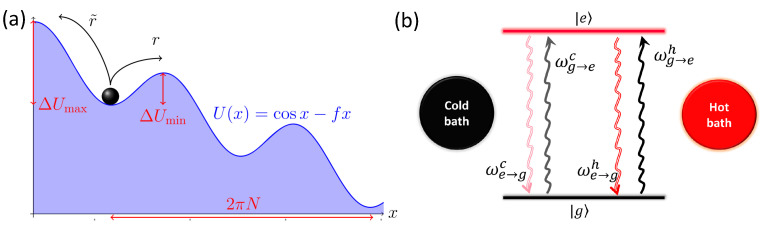
(**a**) The scheme of a particle transport in a periodical skew nonconservative potential U(x)=cos(x)−fx, adopted from [91]; (**b**) sketch of a two-level system connected by two heat baths with the corresponding transfer rate given by $\omega_{i\to j}^{\nu }=e^{-\beta_{\nu }\left(\epsilon_{j}-\epsilon_{i}\right)/2}$ (i,j=g,e, ν=h,c), adopted from [95].

**Figure 18 entropy-24-00813-f018:**
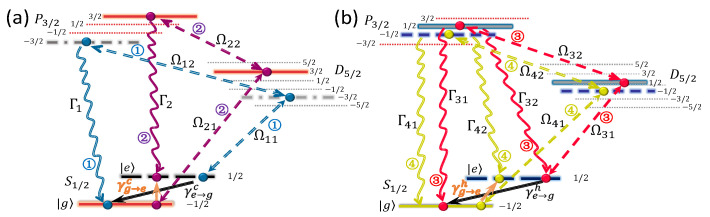
Level schemes to construct effective two-level systems in contact with two heat baths in the trapped ${}^{40}$Ca${}^{+}$ ion, adopted from [95], with the dissipative channels regarding to the cold heat bath (**a**) and hot heat bath (**b**).

**Figure 19 entropy-24-00813-f019:**
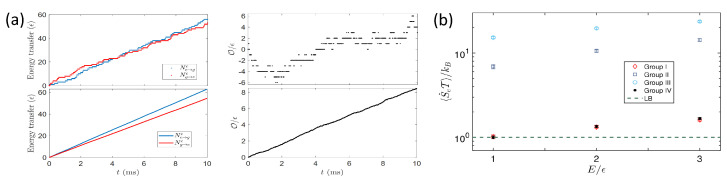
(**a**) Energy transfer (left panels) and net energy flow (right panels) in single stochastic trajectories (top panels) and ensemble average trajectories (bottom panels) regarding the Markovian jumps induced by the cold heat baths, where $N_{e\to g}^{c}$ and $N_{g\to e}^{c}$ correspond, respectively, to the total energy quanta dissipated to the cold baths and the energy quanta transferred into the system from the cold baths. (**b**) Dissipation-time uncertainty relation under different values of the activation energy *E*, where the green dashed line indicates the lower bound, i.e., $k_{B}$. The figures are adopted from [95].

## Data Availability

Not applicable.

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
