# Peer review of "Verification of Information Thermodynamics in a Trapped Ion System"

_entropy, 2022, doi:10.3390/e24060813_

Round 1

Reviewer 1 Report

This manuscript has a potential to become a good review. However the overwhelming amount of language issues makes it very hard to read. The whole review, especially the introduction, should be thoroughly checked by a good English speaking physicist, because just a language correction might, in some cases, alter the physical interpretation off statements. After these corrections are done, the manuscript should be reviewed again. It is not publishable in the present state.

I give just a few examples of problematic language issues (these are jsut examples, the paper is full of such mistakes rendering the text at some places almost impossible to understand):

1) missing words: e.g., "higher temperature than the positive.", "degrees freedom"

2) misleadingly wrong spelling of (key)words: e.g., "efficient temperature" instead of effective temperature, "et. al." instead of etc., "upper operator" instead of raising operator, "temperature of the system is the presentation" ... representation, "the their"

3) capital letters: e.g., "schrödinger" instead of Schrödinger, "dissipative Channel" instead of dissipative channel

4) introduction of terms which are non-standard: e.g., "relative entropy of information theorem" I mean, a system can have entropy, a theorem also but this is probably not what the authors had in mind

5) grammar

Besides, I believe that all explanations in a review article should be as clear as possible so the review can also  be useful to newcomers to the field (at least, all statements should be correct):

1) I would give references where one can find definitions of (anti-)Stokes process, JS model, Kramers relaxation time (some basic textbook is OK)

2) Eq. (10) and below just show Lindblad operators for the discussed situations. These are not Master equations but just parts of RHS of a master equation. A master equation must have time derivative on its LHS...

3) whole sec. 3.2: Please explain why the relaxation in the two-level system is not enough, and one has to enhance it by introducing new relaxation channels.

4) A "reversible dissipative" seems quit contradictory. Usually, processes are either reversible (no entropy production in the universe) or dissipative (entropy production). Please explain.

5) "duration of quantum operators" I think the quantum operator can be applied for some time, but what the duration of quantum operators means is not clear to me...

6) sec. 3.3: It would be good to explain that this bath is about to be connected to the two-level system, right?; "The ground state cooling..." it is not clear why the authors write about this. Please motivate the statements to improve readability...

7) Eqs. (21) and  (22): please write clearly that P_0 corresponds to two-level system and p_i to the bath... Some readers might find this confusing.

8) The description of Maxwell demon at the beginning of 4.1: the authors write the system has uniform temperature, and later they write that the demon can extract work from temperature gradient; the missing part is that the demon can create this gradient by measuring the system...

There are more similar issues later in the text...

Author Response

Thanks a lot for the reviewer’s pertinent comments, and we have tried our best to work on these questions, and made careful modifications based on the reviewer’s comments in the revised manuscript. To clarify some key points, we respond to the comments one-by-one as below. We hope that our response and the paper revision could make the Referee satisfied.

Reviewer 2 Report

The authors make a series of proposals for case studies to study information-thermodynamical relations in a trapped ion system. The subject is interesting and timely; furthermore the authors' study is thorough. Therefore I can recommend publication. A few minor remarks:

-some sloppiness here end there resulting in sentences a bit weirdly formulated. Most striking is the "textTr" on line 359. Please check carefully.
-line 109, what are the "dual natures of quantum and thermodynamics"
-The paper has a lot of content regarding the experimental system, the information thermodynamical bounds and the relation between them. It might be useful for the reader to highlight better the main takeaway points.
-line 187: negative temperature owns higher temperature than the positive? what does this mean?
-eq.(15): it seems that the first term on the RHS represents the Lindblad operators (10). But the Lindblad terms corresponding to the second term of (15) are not defined?

Author Response

Thanks for the reviewer’s pertinent comments and good words for our work. To clarify some key points, we respond to the comments one-by-one. Please see the attachment.

Round 2

Reviewer 1 Report

The authors took all my comments into account and significantly improved the manuscript. However, it still needs a major language revision to become pleasant to read. Some sentences are of the length of a whole paragraph, making them very hard to digest. The authors should at least try to cut the longest of them into several pieces. Otherwise, most readers will simply not be able to read their work. Even the abstract is very dense and immensely difficult to grasp. One example is the sentence 

"Moreover, we elucidate the concrete quantum operators and the decay paths in the trapped-ion system, following a review of quantum state preparation, necessary experimental operations, and effective dissipative processes in the experimental simulation of dynamical processes of thermodynamic models and verification of some newly proposed theories."

It is not wrong, but one needs to almost make notes to decipher it. Why not to make 3 sentences out of it? For example like this (if this is what the authors wanted to say):

"We elucidate the role of concrete quantum operators and the decay paths in the trapped-ion system. We review quantum state preparation and experimental operations necessary to achieve efficient dissipative processes allowing for experimental verification of some newly proposed theories."

The review is still full of such sentences making it also very hard for me to go through the text. 

Besides, some problems with nomenclature are still contained. For example, the phrase "nonequilibrium equation" in the introduction probably stands for "nonequilibrium equality for free energy differences", which generalizes the 2nd law to nonequilibrium situations. Such problems are usually easy to uncover by comparing nomenclature in the manuscript and titles of the cited references. The authors should go again over their review and check the nomenclature carefully.

To conclude, the paper still needs an extensive language revision, preferably from someone who has a glimpse both of English and general physics. This is essential to make the review understandable and thus useful.

Author Response

Thanks a lot for the reviewer’s suggestion, and we have tried our best to revise the manuscript according to the suggestions. Please see the attachment.
